# Hydroarylation of olefins catalysed by a dimeric ytterbium(II) alkyl

Georgia M. Richardson[1], Iskander Douair [2], Scott A. Cameron[3], Joe Bracegirdle [1], Robert A. Keyzers[1], Michael S. Hill[4✉], Laurent Maron [2✉] & Mathew D. Anker [1✉]

Although the nucleophilic alkylation of aromatics has recently been achieved with a variety of potent main group reagents, all of this reactivity is limited to a stoichiometric regime. We now report that the ytterbium(II) hydride, $[BDI^{Dipp}YbH]_2$ ($BDI^{Dipp}$ = CH[C(CH₃)NDipp]₂, Dipp = 2,6-diisopropylphenyl), reacts with ethene and propene to provide the ytterbium(II) $n$-alkyls, $[BDI^{Dipp}YbR]_2$ (R = Et or Pr), both of which alkylate benzene at room temperature. Density functional theory (DFT) calculations indicate that this latter process operates through the nucleophilic ($S_N2$) displacement of hydride, while the resultant regeneration of $[BDI^{Dipp}YbH]_2$ facilitates further reaction with ethene or propene and enables the direct catalytic (anti-Markovnikov) hydroarylation of both alkenes with a benzene C-H bond.

[1] School of Chemical and Physical Sciences, Victoria University of Wellington, Wellington, New Zealand. [2] Université de Toulouse et CNRS, INSA, UPS, UMR 5215, Toulouse, France. [3] Ferrier Research Institute, Victoria University of Wellington, Wellington, New Zealand. [4] Department of Chemistry, University of Bath, Bath, UK. ✉email: msh27@bath.ac.uk; laurent.maron@irsamc.ups-tlse.fr; Mathew.Anker@vuw.ac.nz

The addition of alkyl groups to aromatic systems is a vital transformation in both academic and industrial contexts, providing access to a wide variety of synthetic intermediates, fine chemicals and feedstocks. As benzene does not typically undergo nucleophilic substitution ($S_N1$ or $S_N2$), this is conventionally achieved by the electrophilic aromatic (Friedel–Crafts) substitution of a benzene C–H bond using a strong Lewis acid catalyst (Fig. 1a)[1, 2]. These reactions rely on using reactive alkylating agents such as alkyl halides and, while efficient, their applications are inevitably hampered by their cost and the generation of stoichiometric by-products. In contrast, hydroarylation, the addition of an arene C–H bond across olefins, offers significant advantages over classical Friedel–Crafts alkylation as the reaction is by-product free and the olefin starting materials are generally cheaper and more readily available compared to the corresponding alkyl halide. While a number of potent nucleophilic main group species have very recently been shown to activate and functionalise aryl C–H bonds[3–9], only transition metal complexes have, thus far, provided an alternative catalytic pathway to new alkylated arene products (Fig. 1b)[10–14]. Hydroarylation of olefins catalysed by Ru, Rh, Ir, Pt, Ni, Co and Fe has been extensively studied and, in each case, transition metal alkyl complexes have been identified as catalytically important intermediates which operate via a non-Friedel–Crafts mechanism[14]. These transformations, however, are heavily reliant on directing groups to achieve appreciable levels of site selectivity and suffer from poor specificity toward the formation of branched vs. linear (i.e. Markovnikov vs. anti-Markovnikov) arene products (Fig. 1b)[10–12]. The selective hydroarylation of simple arenes, such as benzene, to linear (anti-Markovnikov) alkyl arene products, therefore, remains a significant challenge.

Despite the first reports of the successful isolation of lanthanide (III) organometallics now dating from ~70 years ago[15–19], there are, hitherto, no reports of their ability to achieve the hydroarylation of olefins. These complexes are, however, highly active for a number of catalytic olefin-based transformations, including cyclization/functionalisation, hydrosilylation, hydroboration, hydrogenation and polymerisation[20, 21]. Although lanthanide(III) alkyls can also facilitate the C–H activation of benzene and other heteroarenes, these processes invariably ensue with the formation of new lanthanide-C(aryl) bonds concomitant with the production of the respective alkane, via a conventional 4-membered σ-bond metathesis transition state (Fig. 1c)[19, 22–25].

In this contribution, we demonstrate the facile synthesis of a highly reactive low-valent and low-coordinate ytterbium(II) hydride, $[BDI^{DiPP}YbH]_2$ ($BDI^{DiPP}$ = CH[C(CH$_3$)NDipp]$_2$, Dipp

= 2,6-diisopropylphenyl), which can sequester both ethene and propene to give the low-coordinate ytterbium(II) n-alkyls [BDI-$^{DiPP}$YbR]$_2$ (R = Et or Pr). Both the ytterbium(II) ethyl and n-propyl derivatives react with either protio- or deuterobenzene at room temperature and density functional theory (DFT) calculations are supportive of a nucleophilic substitution reaction mechanism at benzene reminiscent of $S_N2$ type reactivity. Under select conditions, the ytterbium(II) hydride mediates the exclusively selective, catalytic anti-Markovnikov hydrophenylation of ethene and propene.

## Results and discussion

**Synthesis.** The synthesis of the low-coordinate ytterbium n-alkyls, **3** and **4**, is illustrated in Fig. 2. The reaction of the solvent-free ytterbium bis(amide), $[Yb(N[Si(CH_3)_3]_2)_2]_2$[26], with the β-diimine pro-ligand, $BDI^{DiPP}$-H, refluxed in toluene for 12 h cleanly generated the ytterbium amide, $[BDI^{DiPP}Yb(N[Si(CH_3)_3]_2)]_4$ (**1**) in essentially quantitative yields as a red crystalline solid. The solid-state structure of **1** (Supplementary Fig. 4) is significantly different from the (THF)-coordinated analogue[27] but is closely related to the solvent-free calcium complex, $[BDI^{DiPP}Ca(N[Si(CH_3)_3]_2)]_4$[28]. The addition of phenylsilane to **1** provides the ytterbium hydride, $[BDI^{DiPP}YbH]_2$ **2**, in good yields (>80%) as an extremely air- and moisture-sensitive black crystalline material. In the solid-state, **2** adopts a dimeric structure with two μ$^2$-hydride ligands bridging the Yb(II) centres (Fig. 3a). Each Yb(II) centre binds to the β-diketiminato-N atoms and interacts in a η$^6$-coordination mode (Yb1-Ar$_{cent}$ 2.7099(9) Å) with a Dipp substituent of a second BDI-$^{DiPP}$YbH unit of the dimer. The η$^6$-coordination of **2** is significantly weaker than in other ytterbium(II) complexes with a Yb(II) κ$^1$-N, η$^6$-Dipp chelate (2.424–2.520 Å)[28–30], suggesting that the solid-state structure may not be retained in solution. This geometry contrasts significantly with both of the related dimeric (THF)-coordinated complexes, $[BDI^{DiPP}MH(THF)]_2$ (M = Yb or Ca)[31–33], and the solvent-free calcium analogue, $[BDI^{DiPP}CaH]_2$[6], in which the μ$^2$-hydride ligands provide the sole bridging interactions. When pure isolated samples of **2** are dissolved and analysed by multinuclear nuclear magnetic resonance (NMR) spectroscopy, two distinct species are discriminated in the solution state in a 25:1 ratio. The hydride resonance of the major species displays a significant upfield shift (7.52 ppm, $^1J_{Yb-H}$ = 398 Hz) in comparison to related Yb(II) hydride derivatives such as the (THF)-coordinated, $[BDI^{DiPP}YbH(THF)]_2$ (9.92 ppm, $^1J_{Yb-H}$ = 369 Hz)[31], and the tris(pyrazolyl)borate derivative, $[\{(Tp^{tBu,Me})YbH\}_2]$ ($Tp^{tBu,Me}$ = [HB(3-tBu-5-Me-C$_3$N$_2$)$_3$]$^-$) (10.5 ppm, $^1J_{Yb-H}$ = 369 Hz)[34]. The low frequency of this hydride resonance is, however, closely comparable with that of $[\{[t\text{-BuC(NDipp)}_2]Yb(\mu\text{-}H)\}_2]$ (7.74 ppm, $^1J_{Yb-H}$ = 460 Hz), in which the amidinate ligands are bound to each Yb(II) centre via a κ$^1$-N,η$^6$-Dipp chelate and which display a similar arene interaction[29]. Based on this precedent, therefore, we ascribe the major species observed in the solution spectra of **2** to reflect a similar augmentation of the electron density at Yb and the maintenance of the polyhapto engagement of the aromatic π-system observed in its solid-state structure[35].

In contrast, a second hydride resonance centred at 9.92 ppm ($^1J_{Yb-H}$ = 302 Hz) is associated with the minor component in the solution. This resonance is consistent with other dimeric ytterbium hydrides with only the hydrido ligands μ$^2$-bridging two Yb(II) centres and with no arene interactions. We, therefore, tentatively assign the minor product in solution as **2'** in which both $BDI^{DiPP}$ ligands adopt a symmetrical bidentate N,N-coordination mode to each Yb centre reminiscent of that observed in the calcium complex $[(BDI^{DiPP})CaH]_2$ (Fig. 2)[6]. Although an exchange spectroscopy (EXSY) NMR experiment

**Fig. 1 Aromatic alkylation mechanisms and conventional lanthanide alkyl reactivity with a benzene C–H bond. a** Friedel–Crafts alkylation. **b** Transition metal-catalysed hydroarylation of olefins. **c** σ-bond metathesis of a lanthanide alkyl with a benzene C–H bond.

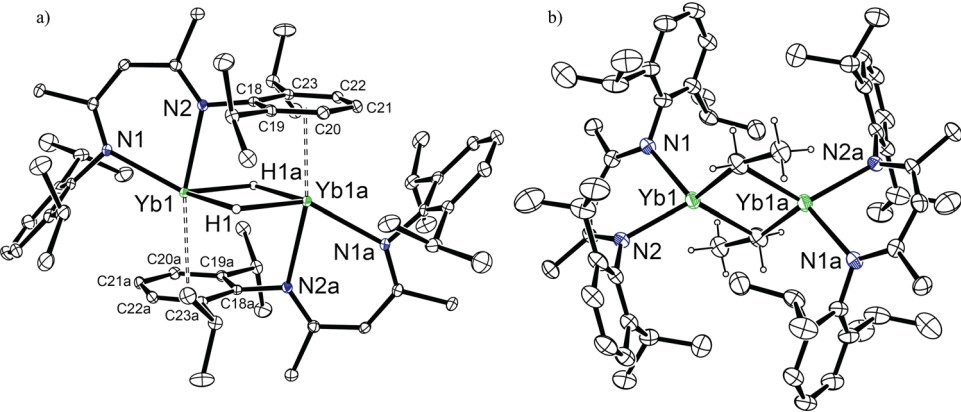

**Fig. 2 Synthesis of compounds 1–4.** Addition of PhSiH₃ to compound **1** provides compound **2**. Reaction of compound **2** with either ethene or propene provides **3** and **4**, respectively.

**Fig. 3 Crystal structures of compounds 2 and 3.** ORTEP representations (30% probability ellipsoids) of (**a**) compound **2**, ($^a = 1 - x, - y, - z$); and (**b**) compound **3** ($^a = 1 - x, - y, - z$). Hydrogen atoms, except the bridging H atoms in **2** and those attached to the α-and β- carbons of **3** have been removed for clarity. Selected bond lengths [Å] and angles [°]: **2**,: Yb1-N1 2.4373(15), Yb1-N1 2.4950(16), Yb1-Cent1 2.7099(9), Yb1-C18$^a$ 3.086, Yb1-C19$^a$ 3.090, Yb1-C20$^a$ 3.018, Yb1-C21$^a$ 3.004, Yb1-C22$^a$ 3.036, Yb1-C23$^a$ 3.081. N1-Yb1-N2 74.05(5), N1-Yb1-Cent1$^a$ 119.44(4), N2-Yb1-Cent1$^a$ 166.38(4). **3**: Yb1-C31 2.502 (11), Yb1-N1 2.328(6), Yb1-N2 2.346(5), Yb1-C31$^a$ 2.557(10), Yb1-C32$^a$ 2.794(9). N1-Yb1-N2 83.28(18), N1-Yb1-C31 111.0(3), N2-Yb1-C31 123.9(2), Yb1-C31-C32 161.3(7).

demonstrated that **2** and **2'** are in equilibrium at room temperature, attempts to quantify this process by variable temperature ¹H NMR studies were frustrated by the competitive redistribution to the homoleptic ytterbium complex, [(BDI$^{Dipp}$)₂Yb][36], and facile Yb–H/D exchange with the deuterobenzene solvent. In this latter regard, the signal associated with the hydride ligand of **2** was observed to decrease if left in benzene-$d_6$ for extended periods of time (>6 h), concomitant with the formation of a new signal in the ²H NMR spectrum centred at 7.92 ppm, which was assigned as the ytterbium deuteride [BDI$^{Dipp}$YbD]₂, **2-$d$**[7, 37].

Exposure of degassed benzene-$d_6$ solutions of **2** to either one atmosphere of ethene or one atmosphere of propene provided the respective ytterbium(II) ethyl (**3**) and n-propyl (**4**) derivatives, as black-brown microcrystalline powders. Crystallisation of **3** from toluene at –30 °C provided black-brown blocks suitable for a single-crystal X-ray diffraction experiment, which demonstrated that **3** crystallises as a centrosymmetric dimer (Fig. 3b). The ytterbium ethyl displays asymmetric ytterbium-to-α-methylene bond lengths (2.503(8), 2.794(9) Å) allowing the identification of the formal intra- and intermolecular Yb–C bonds. The shorter of these Yb–C bond lengths is closely comparable to the terminal Yb (II)-C interactions observed in the tris(trimethylsilyl)-substituted ytterbium dialkyl, [Yb{C(SiMe₃)₃}₂] (2.501(9) Å)[38], but is

significantly shorter than the Yb–C bonds typically observed in benzyl-ytterbium(II) compounds (2.617–2.943 Å)[36, 39]. In addition, each ytterbium centre in compound **3** features an anagostic interaction with two of the hydrogen atoms from the methyl group of the respective ytterbium ethyl. Similarly, compound **4** crystallises as a centrosymmetric dimer (Supplementary Fig. 19), which bears close comparison with **3**, obviating further necessary comment. The isolation of **3** is surprising and in contrast to previous studies that demonstrate that ethene can oxidise ytterbium(II) complexes and act as polymerisation catalysts[40–42].

**Stoichiometric reactivity of 3 and 4 with benzene.** Although the solution integrity of both compounds **3** and **4** was clearly demonstrated by the observation of characteristic high field Yb-α-methylene quartet (**3**: –0.37 ppm) and triplet (**4**: –0.29 ppm) resonances, storage of benzene-$d_6$ solutions of both compounds at room temperature over an extended time period evidenced the complete consumption of the ytterbium alkyl derivatives. This process was concomitant with the formation of the previously identified ytterbium deuteride (**2-$d$**) and the appearance of new multiplet resonances in the ¹H NMR spectra centred at ca. 2.45 ppm (Fig. 4). Encouraged by these observations, a sample of **3** in benzene-$d_6$ was monitored over a 16-hour period by ¹H NMR

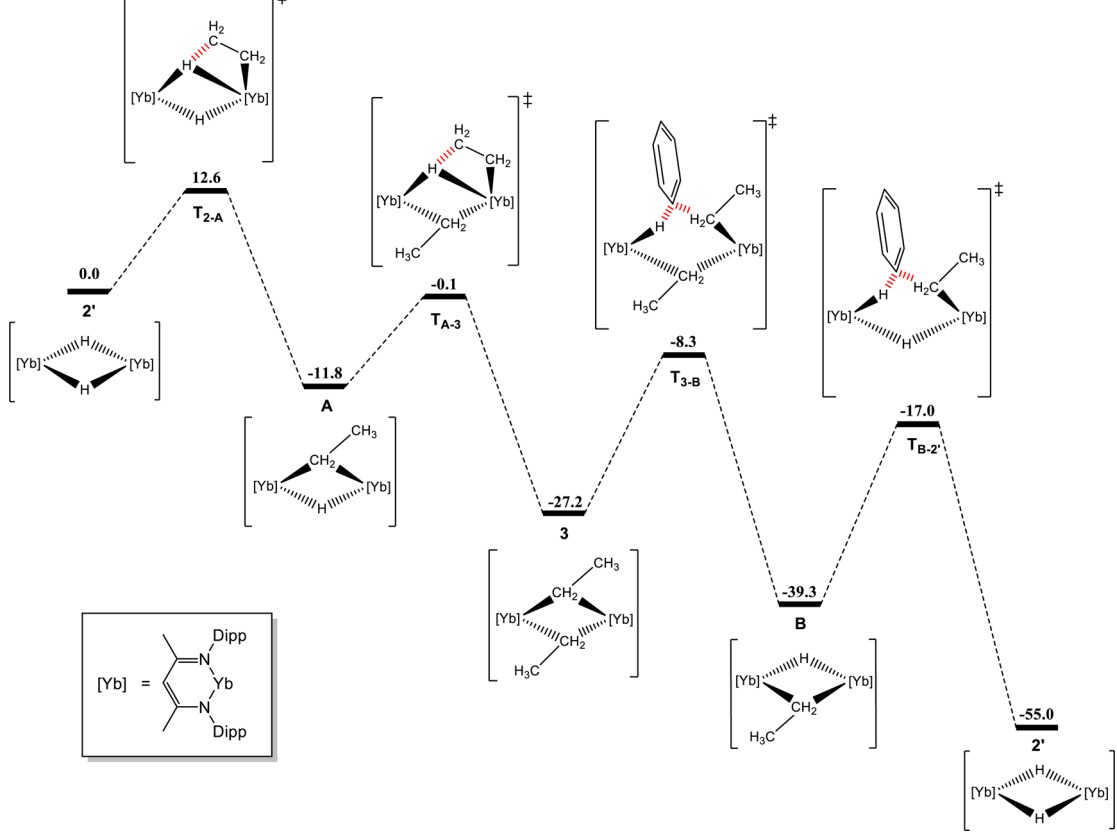

**Fig. 4 Reaction of ytterbium alkyls with benzene.** The addition of benzene to compound **3** at room temperature provides ethylbenzene and the ytterbium deuteride complex **2-d**. The addition of benzene to compound **4** at room temperature provides *n*-propylbenzene and the ytterbium deuteride complex **2-d**.

**Fig. 5 Density functional theory calculations for the reaction of 2 with ethene and benzene.** Computed (DFT, B3PW91) enthalpy profile for the reaction between compound **2** with ethene in benzene at room temperature.

spectroscopy (Supplementary Fig. 20). This reaction resulted in the stoichiometric production of ethylbenzene-$d_5$ identified by $^1$H NMR spectroscopy and gas chromatography-mass spectrometry (GC–MS) as the sole organic product of the reaction. Similarly, analogous treatment of a sample of **4** in benzene-$d_6$ resulted in the stoichiometric production of *n*-propylbenzene-$d_5$ as the sole organic product of the reaction.

**Density functional theory calculations**. To gain greater insight into this previously unobserved reactivity of a lanthanide(II) alkyl, the sequential reactions of the ytterbium hydride **2** with ethene and the resultant ethyl derivative (**3**) with benzene were assessed with DFT (B3PW91) calculations (Fig. 5). Two recent examples of the stoichiometric alkylation of benzene mediated by β-diketiminato heavy alkaline-earth metal analogues have been reported[6, 7]. While both reports confirm a calculated mechanistic pathway that proceeds via the nucleophilic attack at a benzene C–H bond, there are disparities over whether reactivity occurs at a monomeric or dimeric metallic complex[6, 7]. The first system, based on calcium, was suggested to necessitate the initial

dissociation of the dimeric calcium alkyl to a monomeric form before the alkylation of benzene could take place[6]. The second example, based on strontium, questioned the aforementioned monomeric model and suggested that dissociation from a dimer to a charge-separated monomer was unfavourable in the absence of donor solvents. Rather, this study championed a calculated mechanism based on a dimeric alkyl strontium complex that could hinge open to facilitate the nucleophilic reactivity[7]. It is noteworthy that the calculated reaction pathway for our ytterbium system occurs on the bimetallic species, as dissociation of the dimeric hydride **2'** was computed to be significantly endothermic (+25.8 kcal mol$^{-1}$). One molecule of ethene initially inserts into the dimeric ytterbium hydride via the low energy transition state, **T$_{2\text{-}A}$** (+12.6 kcal mol$^{-1}$), to give the diytterbium ethyl-hydride, **A**. Consistent with the facile generation of **3** at room temperature, reaction of this mixed alkyl-hydride with a second molecule of ethene is kinetically accessible via the transition state, **T$_{A\text{-}3}$** (+11.7 kcal mol$^{-1}$). The subsequent barrier towards the first nucleophilic attack of the ethyl-α-methylene carbon towards benzene, via **T$_{3\text{-}B}$**, is slightly higher than the two

insertion transition states ($+18.9$ kcal mol$^{-1}$) but remains accessible at room temperature. At transition state $T_{3\text{-}B}$, the $[Yb_2Et_2]$ dimer hinges open to allow one of the ethyl-α-methylene carbons to attack the carbon atom of a benzene C–H bond, simultaneously enforcing an interaction between the ytterbium and the hydrogen bound to the now 4-coordinate benzene carbon in a process highly reminiscent of $S_N2$ type substitution. Elimination of ethylbenzene is strongly exothermic ($-31$ kcal mol$^{-1}$), regenerating a mixed diytterbium ethyl-hydride, **B**, which can induce a second nucleophilic alkylation of benzene. The barrier towards the second nucleophilic attack of the ethyl-α-methylene carbon towards a further benzene molecule, via $T_{B\text{-}2'}$, is the highest ($+22.3$ kcal mol$^{-1}$) but still accessible at room temperature. At this transition state, $T_{B\text{-}2'}$, and similar to $T_{3\text{-}B}$, the remaining ethyl-α-methylene carbon again attacks the carbon atom of the benzene C–H bond, simultaneously enforcing an interaction between the ytterbium and the hydrogen bound to the now 4-coordinate carbon. The second and final elimination of ethylbenzene is more exothermic ($-38$ kcal mol$^{-1}$) and regenerates the starting dimeric ytterbium hydride **2'**. Although the ability of the dimeric ethyl ytterbium species **3** to polymerise ethene (vide infra) was also investigated by DFT, it is noteworthy that the insertion of ethene into an ytterbium ethyl-α-methylene was found to be 6.1 kcal mol$^{-1}$ less favourable than the nucleophilic alkylation of benzene (Supplementary Fig. 62).

**Catalytic hydrophenylation of ethene and propene.** The net process described by this reactivity represents the stoichiometric hydrophenylation of unactivated terminal alkenes mediated by a lanthanide complex. The ultimate by-product of these reactions is the regeneration of the ytterbium(II) hydride (**2/2'**), the starting material for the synthesis of the ytterbium(II) $n$-alkyl compounds, suggesting that it may be possible to apply this transformation to a catalytic regime. We, therefore, studied the reaction of one atmosphere of ethene in the presence of 0.026 mol/L of ytterbium hydride **2** dissolved in degassed $C_6D_6$ at room temperature. Monitoring of this reaction by $^1$H NMR spectroscopy over the course of 5 days demonstrated a slow but steady decrease in ethene, concomitant with the growth of the diagnostic ethylbenzene methylene signal centred at 2.45 ppm. Visual inspection of the J Young's NMR tubes used to carry out and monitor these reactions, however, also indicated the formation of a precipitate, the identity of which was not apparent in any of the $^1$H NMR spectra (Supplementary Fig. 26). Analysis of the reaction solution demonstrated the presence of ethylbenzene and a small proportion of $n$-butylbenzene in a 100:5 ratio, while the insoluble precipitate was identified as polyethene by analysis with Fourier transform infrared (FTIR) spectroscopy (Supplementary Fig. 35). In an attempt to control the selectivity of this transformation, we repeated the reaction but varied the pressure of ethene across 0.5, 1 and 1.5 atmospheres. No variation in product ratio, however, was observed. While these data confirm that **2** is capable of catalysing the hydroarylation of ethene with benzene, this process is evidently competitive with an oligo/polymerisation process that provides a mixture of linear alkylated benzene and polyethene products. Whilst unselective, this reaction nevertheless represents the first catalytic hydroarylation of an unactivated olefin with benzene mediated by a lanthanide complex (Fig. 6).

Encouraged by these results, we expanded the scope of this catalytic hydrophenylation reactivity to propene. The reaction of one atmosphere of propene in the presence of 0.026 mol/L of ytterbium hydride **2** dissolved in degassed $C_6D_6$ was, therefore, monitored by $^1$H NMR spectroscopy at room temperature over the course of 8 days. This reaction proved to be absolutely selective for the production of $n$-propylbenzene as the sole

organic product of the reaction as identified by multinuclear NMR spectroscopy and GC–MS analysis (Supplementary Figs. 48–56). Although compound **2** is present throughout the early stages of the reaction, it is consumed over the course of 2 hours concomitant with the formation of a new ytterbium alkyl species evidenced by a new upfield triplet resonance centred at $-0.67$ ppm and a new downfield triplet ytterbium hydride resonance (10.35 ppm, $^1J_{Yb\text{-}H} = 251$ Hz), which occur in a 2:1 ratio (Supplementary Fig. 49) and appear simultaneously with the onset of $n$-propylbenzene production. This further ytterbium alkyl species does not correspond to the dimeric $n$-propyl ytterbium derivative **4**, but is tentatively assigned as a mixed ytterbium propyl-hydride complex analogous to species **A/B** included in the calculated reaction pathway (Fig. 5). Analogous alkaline-earth alkyl-hydride species have been implicated as intermediates during the stoichiometric calcium- and strontium-mediated alkylation of benzene[6, 7]. The ytterbium propyl-hydride complex (**5**) is initially present at a low but steady-state concentration in solution but begins to decrease in concentration after 4 days. The production of $n$-propylbenzene ultimately ceased after four catalytic turnovers over the course of 8 days, at which point no spectroscopic evidence could be discerned for the presence of any BDI-coordinated ytterbium species in solution and a visible black precipitate was present in the reaction. The insoluble black precipitate was identified by a single-crystal X-ray diffraction experiment, which disclosed the catalyst breakdown product to be the tetrameric ytterbium(II) allyl complex, $[BDI^{DiPP}Yb(CH_2C(H)CH_2)]_4$ (**6**) (Supplementary Fig. 61). We, therefore, examined the competitive breakdown pathway leading to the formation of **6** by DFT (B3PW91) calculations, compared to the hydroarylation of propene with benzene mediated by **4** (Supplementary Fig. 63). Whilst the global calculated reaction profile is analogous to the hydroarylation of ethene with benzene (Fig. 5) and is significantly exothermic ($-43.4$ kcal mol$^{-1}$), three catalyst deactivation pathways (**C–E**) were identified. Of these, **C** and **E** occur via transition states energetically less favourable ($+1.5$ and $+22.7$ kcal mol$^{-1}$) than the initial insertion of propene into the dimeric ytterbium hydride and the hydroarylation of propene, respectively. However, the reaction of the mixed ytterbium alkyl-hydride with a second molecule of propene to give **4** is disfavoured ($+1.4$ kcal mol$^{-1}$) compared to the deactivation pathway **D** leading to the formation of **6**. This is consistent with the experimentally observed results and rationalises the cessation of catalysis after 8 days. The isolation of an ytterbium(II) allyl complex is reminiscent of a common breakdown pathway reported in transition metal-based olefin hydroarylation catalysis[10], while similar metal allyl formation is a well-precedented observation at single-site group 4 metallocene and post-metallocene catalysts for olefin polymerisation and has even been observed to account for the majority of the metal content in a dormant state under typical propene polymerisation conditions[43–52]. Although notably slow, these results represent a landmark in the reactivity of the lanthanides, mediating the first hydroarylation of olefins using the simplest arene, benzene.

In summary, we have demonstrated the facile synthesis of simple ytterbium(II) hydride complex **2**, which can sequester either ethene or propene to give the respective ethyl or $n$-propyl ytterbium complexes. Both of these $n$-alkyl ytterbium(II) complexes can mediate the hydrophenylation of olefins with benzene. The reactivity of these ytterbium(II) alkyls diverges from all previously reported lanthanide alkyl complexes, which exclusively undergo σ-bond metathesis reactions with aryl C–H bonds, yielding lanthanide aryl and alkane products. DFT calculations suggest that the addition of a benzene C–H bond across double bonds of olefins proceeds via an unusual $S_N2$ type mechanism. The by-product from the stoichiometric reactions of

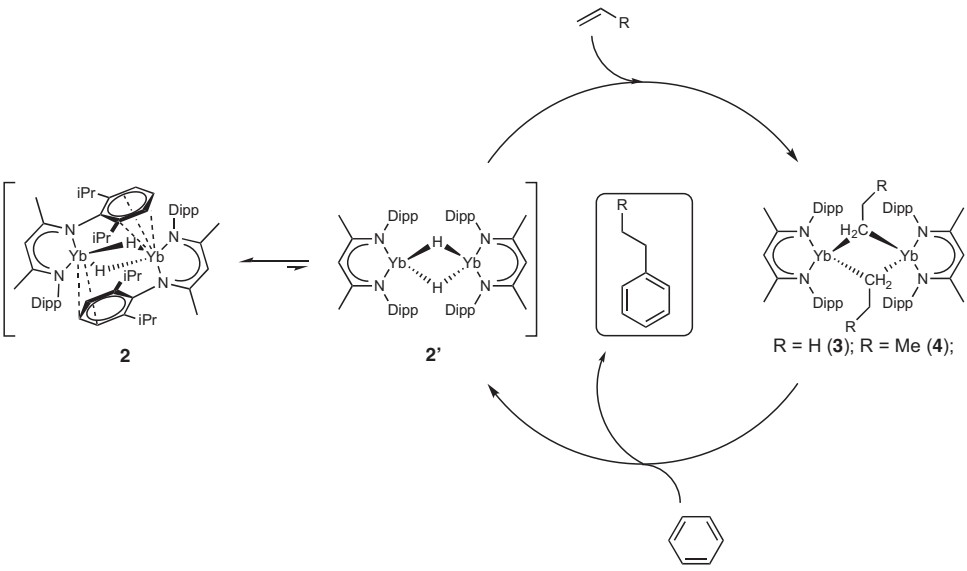

**Fig. 6 Proposed catalytic mechanism.** Hydrophenylation of ethene and propene with benzene catalysed by **3** and **4**, respectively.

the ytterbium *n*-alkyls with benzene is the regeneration of the ytterbium hydride enabling the extension of this reactivity to a catalytic regime. While the hydroarylation of ethene with benzene can be catalysed by **2**, it is unselective with respect to competing for ethene polymerisation. The catalytic hydroarylation of propene with benzene by **2**, however, is completely selective, yielding *n*-propylbenzene as the only organic reaction product, albeit this process is subject to catalyst deactivation through the competitive production of a dormant Yb(II) allyl species. More generally, this study demonstrates that sufficiently nucleophilic lanthanide alkyls have the broader potential to catalyse the hydroarylation of unactivated olefins with arene C–H bonds to provide the linear alkylated arene products.

## Methods

Manipulations were carried out under a dry, oxygen-free argon or dinitrogen atmosphere, with reagents dissolved or suspended in aprotic solvents, and combined or isolated using cannula and glovebox techniques. [BDI$^{Dipp}$Yb(N[Si(CH$_3$)$_3$]$_2$)]$_4$ (**1**) is synthesised by the addition of [Yb(N[Si(CH$_3$)$_3$]$_2$)$_2$]$_2$, to a toluene solution of β-diimine pro-ligand, BDI$^{Dipp}$-H and refluxed in toluene for 12 hours. The reaction of **1** with three equivalents of phenylsilane at room temperature gives [BDI$^{Dipp}$YbH]$_2$ (**2**). [BDI$^{Dipp}$Yb(CH$_2$CH$_3$)]$_2$ (**3**) and [BDI$^{Dipp}$Yb(CH$_2$CH$_2$CH$_3$)]$_2$ (**4**) were synthesised by the addition of the appropriate alkene (ethene for **3** and propene for **4**) to a degassed benzene solution of **2**. All new compounds **1**–**4** and **6** were characterised by elemental analysis, multinuclear NMR spectroscopy and single-crystal X-ray diffraction. Computational details and optimised structures of all intermediates are given. Details are provided in the Supplementary Information.

## Data availability

Crystallographic data for the structures reported in this Article have been deposited at the Cambridge Crystallographic Data Centre under deposition numbers: CCDC 2032031 (**1**), 2032028 (**2**), 2032030 (**3**), 2032029 (**4**), and 2034334 (**6**). These data can be obtained free of charge from The Cambridge Crystallographic Data Centre via www.ccdc.cam.ac.uk/data_request/cif. All data supporting the findings of this study are available within the article, as well as the Supplementary Information file, or available from the corresponding authors on request.

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

## Acknowledgements

We thank the AINSE Early Career Research Grant, Victoria University of Wellington, University Research Fund (M.D.A), AINSE Honours Scholarship, Curtis-Gordon Research Scholarship, Dr. Margaret L. Bailey Award and Victoria University of Wellington Doctoral Scholarship (G.M.R) for financial support for this work. L.M. is a senior member of the Institut Universitaire de France. CalMip is acknowledged for a generous grant of computing time.

## Author contributions

All compounds were synthesised by G.M.R. and M.D.A. and characterised by G.M.R., J.B., R.A.K. and M.D.A. Crystallography was carried out by S.A.C. All computational work was carried out by I.D. under the supervision of L.M. The manuscript was written by M.D.A., M.S.H. and L.M. with contributions from all authors.

## Competing interests

G.R. and M.D.A. are inventors on a provisional patent application No. AU 2021901090. The remaining authors declare no competing interests.
