## [Peer Review File · Nature Communications]

REVIEWER COMMENTS

Reviewer #1 (Remarks to the Author):

Comments:

In this paper, Anker et al report an interesting work on the synthesis of ytterbium(II) n-alkyls, [(BDIDipp)YbR]₂ (R = Et or Pr), which could catalyse the hydrophenylation of ethene and propene to get anti-Markovnikov alkylated aromatic products at room temperature, albeit the turnover is very slow, far away from practical application. Most of the complexes were characterized by multiple methods including NMR, X-ray. DFT calculations were also performed to investigate the potential process. In 2017, the group of Hill reported a groundbreaking result about the nucleophilic alkylation of benzene mediated by calcium complexes (Science, 2017, 358, 1168-1171). Soon after, the group of Harder found that the heavier alkaline earth metal (Sr) complex [(BDIDIPeP)SrEt]₂ could also prompt the similar transformations (Angew. Chem. Int. Ed., 2019, 58, 5396-5401). The reactivity of strontium alkyl [(BDIDIPeP)SrEt]₂ toward ethylene polymerization and the catalytic nucleophilic aromatic substitution with benzene was demonstrated as well. Due to the similar ionic radii (Yb²⁺, 1.02 Å; Ca²⁺, 1.00 Å; CN = 6), the similar structure and chemistry of Ca²⁺ and Yb²⁺ has been recognized for a long time. Thus, the aromatic substitution of benzene mediated by Yb²⁺ complex [(BDIDipp)YbR]₂ can be regarded as the extended work of the previous calcium chemistry. The following issues and mistakes should be addressed before consideration for acceptance in Nature Communications.

Major issues:

1. The scope of olefins is limited. In the previous work, Hill et al disclosed the nucleophilic alkylation of benzene with a range of olefins, including ethene, 1-butene, and 1-hexene. In this work, the authors mentioned that the catalytic hydroarylation of propene with benzene could be terminated by dormant Yb(II) allyl species (6). Whether this termination will happen to other olefins? Beside alpha olefins, is intern alkenes can be used as the olefin substrate? Thus, more olefin substrates should be tested.
2. Yb(II) allyl complex (6) was characterized only by X-ray analysis, without isolated yield and NMR data. Due to the high disorder of allyl group in the crystal structure, NMR data should be provided to corroborate the structure.
3. The isolation of Yb(II) allyl complex (6) indicates that there are two competitive routes. One is the addition of benzene C-H bond across propene, which leads to n-propyl-benzene. The other is the dehydrogenation of Yb-H with propene, which leads to irreversible formation of the tetrameric Yb(II) allyl complex, [BDIDippYb(CH₂C(H)CH₂)₄]. DFT calculation should provide the energy difference between these two routes.
4. In lines 197-205, the authors state that there is only diytterbium n-propyl-hydride (5), which is at low and stable concentration in beginning, with the absence of dimeric n-propyl ytterbium species. This observation is in contrast to the previous Harder's report of strontium-mediated alkylation of benzene, in which [Sr₂(Et)H] and [SrEt]₂ exist simultaneously that decrease gradually with the reaction time extending. What is the potential reason for this difference?
5. For clarity, all the H atoms, except for the bridging hydride (H1) and the hydrogen atom in bridging Et or Pr groups, are omitted. The ORTEP of compound 2, 3 and 4 should be re-drawn with the labels of coordinated aryl ring.
6. Can other benzene derivatives be activated, such as naphthalene and anthracene.

Minor issues:

1. ¹⁷¹Yb NMR is indispensable, especially for Yb-H/D complexes.
2. Polyethylene (or oligomer) was characterized only by FTIR, the Mn and PDI should also be provided.
3. The agnostic interactions of bridging Et or Pr with Yb center should be mentioned.
3. In the supporting data, two CIF file are duplicated. The CIF of complex 2 (Yb-H) is missing. According to the requirement of CCDC deposition, the HKL, res and ins information should be included in new CIF files.
4. Page S65-S66, largest diff. peak and hole were found in complex 1, 3 and 6, please provide the explanations.

- Page S64, please provide more details about the refinement of complexes, especially the location of bridging hydride in complex 2, the disordered Yb atom in complex 4, and ligand disorder in other complexes.
- Page S26, key bond distances of complex 4 should be listed.

Too many typos and careless mistakes were found in this paper:

- Line 1: "an Ytterbium(II) Alkyl " change to "two Ytterbium(II) Alkyls"
- Line 23: should be "both academic and industrial".
- Line 56: change "ytterbium(II) n-alkyls [BDIDippYR]₂" to "ytterbium(II) n-alkyls [(BDIDipp)YbR]₂".
- Line 80-81, the coupling constant should be listed in [(BDIDipp)YbH(THF)]₂ (9.92 ppm, J_{Yb-H} = ? Hz) and [(TptBu,Me)YbH]₂ (10.5 ppm, J_{Yb-H} = ? Hz).
- Line 81: "[HB(2-Me,3-tBu-C₃N₂)₃]- " should be "[HB(3-tBu-5-Me -C₃N₂)₃]- ".
- Line 118, Yb1-C31 2.03(8)? It is impossible!!
- Line 137: "heavy Grignard analogues" change to "heavy alkaline-earth metals analogues" may be preferable.
- Line 196: "in a 1:1 ratio (Figure S49)", should be 2:1.
- Line 222: " the addition of benzene C-H bond across our ytterbium n-alkyls proceeds via an unusual SN₂ type mechanism." change to " the addition of benzene C-H bond across double bonds of olefins proceeds via an unusual SN₂ type mechanism."
- Lines 275: the journal name "science" is missing.
- Lines 297, 300, 422: the references are not in the right format, and the similar mistakes should be checked.
- Lines 301: delete the last "n" in the "Angew. Chem. Int. Ed.n,".
- In Figure 2 and 6, check carefully about the R!
- Figure S13 and S20 are twisted.
and more...

Reviewer #2 (Remarks to the Author):

Hill, Maron, Anker and coworkers report the catalytic hydroarylation of olefins with an ytterbium(II) alkyl catalyst. The proposed SN₂-like mechanism is quite different from the more common sigma-bond metathesis that occurs with many lanthanide molecules. While I think this report opens the door to many questions, particularly towards how changes to the lanthanide complex (ligand and metal) will alter how favorable this new process is, I think this communication is important to the lanthanide and catalysis communities and is suitable for publication in Nature Communications. I think a few minor revisions should be addressed prior to publication, as listed below.

- In Figure 2, it is indicated that 2' is the structure that shows insertion with an olefin to form 3 or 4. While in Figure 4, complex 2-d is indicated to form before equilibrium is established between 2-d and 2'-d. I don't think this report shows any conclusive evidence of for these specific details. I would suggest putting complexes 2 and 2' in equilibrium in brackets. Same for Figure 6.
- For the use of 3 for catalysis, can you alter the ratio of production of n-ethyl-benzene, n-butylbenzene and polyethylene by varying the pressure of ethylene provided?
- I believe complexes 3 and 4 are misdrawn in the Figures. The alkyls should be CH₂-CH₂R, where R = H or Me. Additionally, the product in the box on Figure 6 is missing a CH₂ as well.
- Since the NMR's and GCMS studies are critical to these results, I think the figures in the Supplementary Material should be clearer. As it stands, readers who want to understand the chemistry at a deeper level than the communication itself will need to put too much effort into understanding the results than they should. For example:
-In some spectra only some peaks are integrated, while others are left. In some cases these may be byproducts or solvents, but it should be noted.

- In some cases there are multiple things in solution, and chemdraws are provided, but there is no labelling to indicate which peaks correspond to which product.
- For the catalysis NMR arrays, I suggest that the key peaks being consumed and growing in should be noted in some way
- Some labels are too small to read. I encourage the authors to increase the size of the integrations and x-axis of most NMR plots in the Supplementary Material. The same is true for GCMS traces

Reviewers' Comments to the Authors:

Reviewer 1

Major issues:

1. The scope of olefins is limited. In the previous work, Hill et al disclosed the nucleophilic alkylation of benzene with a range of olefins, including ethene, 1-butene, and 1-hexene. In this work, the authors mentioned that the catalytic hydroarylation of propene with benzene could be terminated by dormant Yb(II) allyl species (6). Whether this termination will happen to other olefins? Beside alpha olefins, is intern alkenes can be used as the olefin substrate? Thus, more olefin substrates should be tested.

Author Response: While we appreciate the reviewers feedback, we respectfully disagree. While the previous work by Hill et al. only included an additional two olefins, 1-butene and 1-hexene, the report was limited to a stoichiometric regime and no catalytic reactivity was achieved. In our communication, we move beyond this stoichiometric reactivity and demonstrate that our Yb(II) alkyls can facilitate the first example of hydroarylation of both ethene and propene with the unactivated arene, benzene. We refer to Reviewer 2's comment that "While I think this report opens the door to many questions, particularly towards how changes to the lanthanide complex (ligand and metal) will alter how favourable this new process is, I think this communication is important to the lanthanide and catalysis communities". However, as suggested by Reviewer 1, during our revisions we studied the hydroarylation of 1-butene but found it was limited to a strictly stoichiometric regime due to a related deactivation pathway that we have included in the revised manuscript. This conclusion demonstrates that a spectator ligand redesign will be necessary to take this study beyond the initial observation of the first example of hydroarylation of olefins by a lanthanide complex, and would therefore constitute a completely separate study.

2. Yb(II) allyl complex (6) was characterized only by X-ray analysis, without isolated yield and NMR data. Due to the high disorder of allyl group in the crystal structure, NMR data should be provided to corroborate the structure.

Author Response: Thank you for this suggestion. However, due to the extreme insolubility of the Yb(II) allyl complex (6) in hydrocarbon solvents (benzene, toluene, cyclohexane methylcyclohexane) even at elevated temperatures and compound degradation in coordinating solvents (THF, Et₂O, Py), we have been unable to obtain NMR data. We have added this limitation on Page: S69 in the supplementary.

" Due to the extreme insolubility of the Yb(II) allyl complex 6 in hydrocarbon solvents (benzene, toluene, cyclohexane methylcyclohexane) even at elevated temperatures and compound degradation in coordinating solvents (THF, Et₂O, Py), we have been unable to obtain accurate NMR spectroscopic data."

3. The isolation of Yb(II) allyl complex (**6**) indicates that there are two competitive routes. One is the addition of benzene C-H bond across propene, which leads to n-propylbenzene. The other is the dehydrogenation of Yb-H with propene, which leads to irreversible formation of the tetrameric Yb(II) allyl complex, [BDIDippYb(CH₂C(H)CH₂)]₄. DFT calculation should provide the energy difference between these two routes.

Author Response: As suggested by the reviewer, we have performed the DFT calculations that provide the energy difference between the two competing routes. The revised text reads as follows on line 213 - 221:

“ We therefore examined the competitive breakdown pathway leading to the formation of **6** by DFT (B3PW91) calculations, compared to the hydroarylation of propene with benzene mediated by **4** (Figure S63). Whilst the global calculated reaction profile is analogous to the hydroarylation of ethene with benzene (Figure 5) and is significantly exothermic (–43.4 kcal mol⁻¹), three catalyst deactivation pathways (**C** – **E**) were identified. Of these, **C** and **E** occur *via* transition states energetically less favourable (+1.5 and +22.7 kcal mol⁻¹) than the initial insertion of propene into the dimeric ytterbium hydride and the hydroarylation of propene respectively. However, the reaction of the mixed ytterbium alkyl-hydride with a second molecule of propene to give **4** is disfavoured (+1.4 kcal mol⁻¹) compared to the deactivation pathway **D**, leading to the formation of **6**. This is consistent with the experimentally observed results and rationalises the cessation of catalysis after 8 days.”

4. In lines 197-205, the authors state that there is only ditytterbium n-propyl-hydride (**5**), which is at low and stable concentration in beginning, with the absence of dimeric n-propyl ytterbium species. This observation is in contrast to the previous Harder's report of strontium-mediated alkylation of benzene, in which [Sr₂(Et)H] and [SrEt]₂ exist simultaneously that decrease gradually with the reaction time extending. What is the potential reason for this difference?

Author Response: We thank the reviewer for posing this question. Harder's report focused on the reactivity of the group 2 element strontium, and due to the larger ionic radii of this element (Sr²⁺, 1.32 Å), used the significantly bulkier BDI^{Dipep} ligand (Angew. Chem. Int. Ed., 2019, 58, 5396-5401). In this report, we utilise the significantly smaller lanthanide ytterbium ion (Yb²⁺, 1.02 Å) and smaller BDI^{Dipp} ligand. Therefore, it is impossible to directly compare these compounds with any meaningful level of accuracy, but we speculate that they are likely to have significantly different thermodynamic minima in solution which could lead to this divergence in observed reactivity. Although we agree that this is an interesting question, for the reasons above, it is not appropriate for inclusion in this manuscript.

5. For clarity, all the H atoms, except for the bridging hydride (H1) and the hydrogen atom in bridging Et or Pr groups, are omitted. The ORTEP of compound **2**, **3** and **4** should be re-drawn with the labels of coordinated aryl ring.

Author Response: Thank you for this suggestion. We have redrawn the ORTEP of compounds **2**, **3**, **4** and **6** with all hydrogen atoms omitted except for the bridging H atoms in **2** and the bridging Et and Pr groups in **3** and **4**. Figure 3 (Line: 116) has been updated in the manuscript. We have included comments (line 118 – 119):

“Hydrogen atoms, except the bridging H atoms in **2** and those attached to the α - and β - carbons of **3** have been removed for clarity.”

6. Can other benzene derivatives be activated, such as naphthalene and anthracene.

Author Response: Thank you for pointing this out. Although we agree that this is an important consideration, it is beyond the scope of this manuscript. In this communication, we demonstrate and aim to rapidly communicate that our ytterbium(II) alkyls can catalyse the hydroarylation of ethene and propene with benzene. As a non-activated arene, achieving this transformation with benzene represents the pinnacle of this new reactivity.

Minor issues:

1. ^{171}Yb NMR is indispensable, especially for Yb-H/D complexes.

Author Response: Thank for this suggestion. However, after multiple attempts we have not been able to locate the ^{171}Yb NMR signal for either of these compounds. Whilst ^{171}Yb NMR is an indispensable technique, ytterbium hydride compounds are regularly reported in the literature without the ^{171}Yb NMR spectrum because of the difficulty in locating the signal.

For example:

Ruspic, C., Spielmann, J., Harder, S. Syntheses and structures of ytterbium(II) hydride and hydroxide complexes: similarities and differences with their calcium analogues. *Inorg. Chem.* 2007, **46**, 5320-5326.

Basalov, I. V., Lyubov, D. M., Fukin, G. K., Shavyrin, A. S., Trifonov, A. A. A double addition of Ln-H to a carbon-carbon triple bond and competitive oxidation of ytterbium(II) and hydrido centers. *Angew. Chem. Int. Ed.* 2012, **51**, 3444-3447.

Yan, K., Schoendorff, G., Brianna, M., U., Ellern, A., Windus, T. L., Sadow, A., D. Intermolecular beta-hydrogen abstraction in ytterbium, calcium and potassium tris(dimethylsilyl)methyl compounds. *Organometallics* 2013, **32**, 1300 – 1316.

2. Polyethylene (or oligomer) was characterized only by FTIR, the Mn and PDI should also be provided.

Author Response: We would like to highlight to the reviewer that polyethene contains no heteroatom or double bonds for possible proton or metal ion attachment, presenting a problem for ionization, excluding common mass spectrometry techniques. Polyethene is also insoluble

in common solvents, excluding NMR spectroscopic analysis. The conventional and most widely used technique to analyse the properties of polyethene is FTIR-spectroscopy (please see references below). We have already provided the FTIR of our sample of polyethene and have added these references to the supplementary information.

Gulmine, J. V., Janissek, P. R., Heise, H. M. & Akcelrud, L. Polyethylene characterization by FTIR. *Polymer Testing* **21**, 557-563, doi:[https://doi.org/10.1016/S0142-9418\(01\)00124-6](https://doi.org/10.1016/S0142-9418(01)00124-6) (2002).

Kovács, R. L. *et al.* Surface characterization of plasma-modified low density polyethylene by attenuated total reflectance fourier-transform infrared (ATR-FTIR) spectroscopy combined with chemometrics. *Polymer Testing* **96**, 107080, doi:<https://doi.org/10.1016/j.polymertesting.2021.107080> (2021).

Spina, F. *et al.* Low density polyethylene degradation by filamentous fungi. *Environmental Pollution* **274**, 116548, doi:<https://doi.org/10.1016/j.envpol.2021.116548> (2021).

Kelly, K., Brown, G. & Anthony, S. Quantifying CTFE content in FK-800 using ATR-FTIR and time to peak crystallization. *International Journal of Polymer Analysis and Characterization* **25**, 621-633, doi:[10.1080/1023666X.2020.1827859](https://doi.org/10.1080/1023666X.2020.1827859) (2020).

Suman, T. Y. *et al.* Characterization of petroleum-based plastics and their absorbed trace metals from the sediments of the Marina Beach in Chennai, India. *Environmental Sciences Europe* **32**, 110, doi:[10.1186/s12302-020-00388-5](https://doi.org/10.1186/s12302-020-00388-5) (2020).

3. The anagostic interactions of bridging Et or Pr with Yb center should be mentioned.

Author Response: We have included comments on the anagostic interactions (line 110 – 112): “Additionally, each ytterbium centre in compound **3** features an anagostic interaction with two of the hydrogen atoms from the methyl group of the respective ytterbium ethyl.”

4. In the supporting data, two CIF file are duplicated. The CIF of complex 2 (Yb-H) is missing. According to the requirement of CCDC deposition, the HKL, res and ins information should be included in new CIF files.

Author Response: Thank you for pointing this out. We now have all HKL and RES information within the CIF files for all structures.

5. Page S65-S66, largest diff. peak and hole were found in complex 1, 3 and 6, please provide the explanations.

Author Response: We now have discussed and provided explanations for methods used in the data analysis of all crystallographic data. This can be found on Page: S64 – S67 in the supplementary.

6. Page S64, please provide more details about the refinement of complexes, especially the location of bridging hydride in complex 2, the disordered Yb atom in complex 4, and ligand disorder in other complexes.

Author Response: Please see response to point 5.

7. Page S26, key bond distances of complex 4 should be listed.

Author Response: Key bond distances for complex 4 have been listed on page: S26 of the supplementary.

8. Line 1: “an Ytterbium(II) Alkyl ” change to “two Ytterbium(II) Alkyls”

Author Response: We think this is a good suggestion but for accuracy have changed the title of the paper to (Line: 1).

“Hydroarylation of Olefins Catalysed by a Dimeric Ytterbium(II) Alkyl”

9. Line 23: should be “both academic and industrial”.

Author Response: We thank the reviewer for this observation but in order for the sentence to make sense we have changed it to (Line: 23):

“The addition of alkyl groups to aromatic systems is a vital transformation in both academic and industrial contexts, providing access to a wide variety of synthetic intermediates, fine chemicals and feedstocks.”

10. Line 56: change “ytterbium(II) n-alkyls [BDIDippYR]₂” to “ytterbium(II) n-alkyls [(BDIDipp)YbR]₂”.

Author Response: We have changed this to (Line: 55):

“ytterbium(II) n-alkyls [(BDI^{Dipp})YbR]₂”

11. Line 80-81, the coupling constant should be listed in [(BDIDipp)YbH(THF)]₂ (9.92 ppm, J_{Yb-H} = ? Hz) and [(TptBu,Me)YbH]₂ (10.5 ppm, J_{Yb-H} = ? Hz).

Author Response: We have changed this to include the coupling constants (Line: 80 – 82): “[(BDI^{Dipp})YbH(THF)]₂ (9.92 ppm, ¹J_{Yb-H} = 369 Hz),³¹ and the tris(pyrazolyl)borate derivative, [(Tp^{tBu,Me})YbH]₂ (Tp^{tBu,Me} = [HB(2-Me,3-*t*Bu-C₃N₂)₃]⁻) (10.5 ppm, ¹J_{Yb-H} = 369 Hz).”

12. Line 81: “[HB(2-Me,3-*t*Bu-C₃N₂)₃]⁻ ” should be “[HB(3-*t*Bu-5-Me -C₃N₂)₃]⁻ ”.

Author Response: We have changed this to (Line: 81):

“(Tp^{tBu,Me} = [HB(3-*t*Bu-5-Me-C₃N₂)₃]⁻)”

13. Line 118, Yb1-C31 2.03(8)? It is impossible!!

Author Response: Thank you for pointing this out, we have corrected this to (Line: 122):
“Yb1-C31 2.502(11)”

14. Line 137: “heavy Grignard analogues” change to “heavy alkaline-earth metals analogues” may be preferable.

Author Response: We have changed the text as suggested to (Line: 141):
“heavy alkaline-earth metal analogues”

15. Line 196: “in a 1:1 ratio (Figure S49)”, should be 2:1.

Author Response: We have changed the text as suggested to (Line: 202):
“2:1”

16. Line 222: “ the addition of benzene C-H bond across our ytterbium n-alkyls proceeds via an unusual SN2 type mechanism.” change to “ the addition of benzene C-H bond across double bonds of olefins proceeds via an unusual SN2 type mechanism.”

Author Response: We have changed the text as suggested to (Line 235):
“the addition of benzene C-H bond across double bonds of olefins proceeds via an unusual SN2 type mechanism.”

17. Lines 275: the journal name “science” is missing.
18. Lines 297, 300, 422: the references are not in the right format, and the similar mistakes should be checked.
19. Lines 301: delete the last “n” in the “Angew. Chem. Int. Ed.n.”.

Author Response: Thank you for pointing these out. We have carefully gone over all the references to ensure that they are all in the correct format (Line: 274 – 448).

20. In Figure 2 and 6, check carefully about the R!

Author Response: Please see response 3 to reviewer 2.

21. Figure S13 and S20 are twisted.

Author Response: We have made changed these to landscape images.

Reviewer 2

Hill, Maron, Anker and co-workers report the catalytic hydroarylation of olefins with an ytterbium(II) alkyl catalyst. The proposed SN2-like mechanism is quite different from the more

common sigma-bond metathesis that occurs with many lanthanide molecules. While I think this report opens the door to many questions, particularly towards how changes to the lanthanide complex (ligand and metal) will alter how favourable this new process is, I think this communication is important to the lanthanide and catalysis communities and is suitable for publication in Nature Communications. I think a few minor revisions should be addressed prior to publication, as listed below.

Author Response: We thank Reviewer 2 for their recognition that this report is important to both the lanthanide and catalysis communities, and for their view that this report is suitable for publication in Nature Communications.

1. In Figure 2, it is indicated that 2' is the structure that shows insertion with an olefin to form 3 or 4. While in Figure 4, complex 2-d is indicated to form before equilibrium is established between 2-d and 2'-d. I don't think this report shows any conclusive evidence of for these specific details. I would suggest putting complexes 2 and 2' in equilibrium in brackets. Same for Figure 6.

Author Response: We agree with the reviewers assessment. Accordingly, Figure 2 (Line 88), Figure 4 (Line 136) and Figure 6 (Line 192) have been edited to include the equilibrium of 2 and 2' in brackets.

2. For the use of 3 for catalysis, can you alter the ratio of production of n-ethyl-benzene, n-butylbenzene and polyethylene by varying the pressure of ethylene provided?

Author Response: As suggested by the reviewer we have varied the pressure of ethene, .5, 1 and 1.5 atmospheres (the upper limit of our apparatus) to in an attempt to alter the product ratio of n-ethyl-benzene, n-butylbenzene and polyethylene but these pressure differences had no effect on the ratios observed. We have added this comment to the manuscript (Line 186 – 188):

“In an attempt to control the selectivity of this transformation, we repeated the reaction but varied the pressure of ethene across 0.5, 1 and 1.5 atmospheres. No variation in product ratio, however, was observed.”

3. I believe complexes 3 and 4 are misdrawn in the Figures. The alkyls should be CH₂-CH₂R, where R = H or Me. Additionally, the product in the box on Figure 6 is missing a CH₂ as well.

Author Response: Thank you for pointing this out. We have amended Figure 2 (Line 88), Figure 4 (Line 136) and Figure 6 (Line 192) to include R= H (3) and R = Me (4) and added the CH₂ to the product in the box.

4. Since the NMR's and GCMS studies are critical to these results, I think the figures in the Supplementary Material should be clearer. As it stands, readers who want to understand the chemistry at a deeper level than the communication itself will need to put too much effort into understanding the results than they should. For example:

-In some spectra only some peaks are integrated, while others are left. In some cases these may be by-products or solvents, but it should be noted.

- In some cases there are multiple things in solution, and chemdraws are provided, but there is no labelling to indicate which peaks correspond to which product.

Author Response: We have added chemdraws to the NMR spectra in the supplementary (Page: S34, S35, S43, S44, S55, S56) and labeled the peaks which correspond to a particular product.

- For the catalysis NMR arrays, I suggest that the key peaks being consumed and growing in should be noted in some way

Author Response: We have added chemdraws to the NMR spectra in the supplementary (Page: S28, S31, S33, S42, S52, S53, S62 and S63) and highlighted the peaks which correspond to a particular product.

- Some labels are too small to read. I encourage the authors to increase the size of the integrations and x-axis of most NMR plots in the Supplementary Material. The same is true for GCMS traces

Author Response: Thank you for pointing this out. We have edited all of the NMR spectra and GCMS traces so that all axis text and integrations have increased in text size to be easily read.

REVIEWERS' COMMENTS

Reviewer #1 (Remarks to the Author):

Most of the concerns have been well addressed. I agree publication as current form.

Reviewer #2 (Remarks to the Author):

The authors have addressed my concerns and I now believe that it is suitable for publication in Nature Communications